# Pre-Service Teachers' Critical Digital Literacy Skills and Attitudes to Address Social Problems

**Jordi Castellví** [1],*, **María-Consuelo Díez-Bedmar** [2] and **Antoni Santisteban** [1]



1   Department of Language, Literature and Social Sciences Education, Faculty of Education, Autonomous University of Barcelona, Bellaterra, 08193 Cerdanyola del Vallès, Spain; antoni.santisteban@uab.cat
2   Department of Science Education, Faculty of Humanities and Educational Sciences, University of Jaen, 23071 Jaén, Spain; mcdiez@ujaen.es
*   Correspondence: jordi.castellvi.mata@uab.cat

**Abstract:** The emergence and expansion of social networks in the digital age has led to social transformations that have a great impact within the field of education. Teacher-training programs face the challenge of preparing future teachers to critically interpret digital media. They must succeed in this if we are to develop citizens who are well informed and reflective, which then raises the question: Are future teachers critical thinkers? This study took third- and fourth-year students of primary education (n = 322) at five Spanish universities and explored their capacity for constructing critical discourses. It examined how well they can analyze and discuss information from digital media on social problems like poverty, economic crises, social justice, and the media. Its findings reveal that future teachers have difficulty in putting together critical discourses based on information from the Internet on social problems. Those who have doubts, compare, analyze, and reason are the minority.

**Keywords:** critical digital literacy; social problems; pre-service teachers

## 1. Introduction

### 1.1. Learning in a Digital Age

The digital age, with its apparently limitless worldwide social, cultural, and economic exchanges (Pérez-Gómez 2012), has changed the way we interact through the media and how we access information (Bauman 2008). Not only that, it has also influenced all communication, information, and learning areas, whether they are digital or not (Castellví et al. 2020b). Social and digital media have generated reading and writing practices, which points to a cultural change of prime importance (Pérez-Tornero 2011), one which has not been accompanied by an educational change to help in selecting, discriminating, or evaluating the information in the media.

The Internet, thanks to its immediacy and accessibility, has become one of the primary sources of information for young people (Blikstad-Balas 2015; Castellví 2019). However, the information it offers us often reproduces hegemonic narratives and can be lacking in rigor, or even biased or false. On the other hand, digital and social media can play a crucial role in fostering democratic participation, social and civic engagement, and social change (Díez-Gutiérrez and Díaz-Nafría 2018; Kellner and Share 2007) and can be a powerful tool for countering hegemonic narratives.

### 1.2. Critical Digital Literacy Skills and Attitudes

One of the most important authors of critical pedagogy is Paulo Freire. With critical pedagogy, Freire set the basis of what we understand by critical thinking. Freire reconstructed what it means

to be a teacher, showing that education should not be a technique or a list of steps to follow but a transformative action. Freire and Macedo (2004) also introduced a new meaning for the word "literacy". Freire understood that through literacy, and through developing critical thinking, people can change themselves and change society, in a process of liberation. For Freire, the ability to use the written word is essential to transforming the world, and it makes sense when its learning leads us to this praxis. Thus, according to Freire, literacy would be an attitude, rather than a mere skill.

In the early 1990s, with ICT still in its infancy, Lankshear and McLaren (1993) published a work that for the first time took critical literacy as its central theme. Influenced by Freire's thesis of reading the word to read the world (Freire and Macedo 2004), they tried to respond to the propositions made in the preceding decades regarding the formation of critical thinking, which they saw as overly cognitivist. They argued that critical thinking is not merely a set of skills but also an attitude to information, a way of thinking and living, which should prepare us for action and social transformation.

Critical digital literacy (CDL) is recent as a field of study having emerged from critical literacy and critical media literacy, among other fields. CDL is not only about developing technology or critical thinking skills but is a preparation for living in a digital world. Hence, we consider CDL to be the skills and attitudes needed to search for information, analyze multiple multimodal texts, reflect on information, create narratives and counter-narratives, and act socially on the pursuit of radical democracy and global justice (Castellví 2019; Castellví et al. 2020a; Kellner and Share 2007; McDougall et al. 2018; Santisteban et al. 2020).

## 2. Educating Critical Teachers

### 2.1. The Digital World Is Challenging

The idea put forward by Freire and Macedo (2004)—to read the word and the world in order to produce citizens who are literate and therefore able to act in their society—today takes on a very important meaning.

The impact of platforms like YouTube on adolescents who have just completed their primary education (García-Jiménez et al. 2017), or how young people are seen to react to cultural representations of the resistance through Twitter (Torrego and Gutierrez 2016), confirm the conclusion drawn by Ramírez and González (2016), when they warn that the development of media competence, in all its dimensions, depends on the teachers' flexibility in interpreting the curriculum—an ability that depends, in turn, on the training received.

The cultural changes brought by the digital age place us in a digital world that is challenging for schools. It is a matter of urgency for teachers to be trained in CDL (Garrett et al. 2020; McDougall et al. 2018; Meehan et al. 2015; Santisteban et al. 2020; Schwartz 2001) so that they can design educational interventions and develop alternative curricular materials, going beyond the mere use of technology as a teaching resource and holding texts and narratives up to scrutiny from all angles.

Studies like the one conducted by Alonso et al. (2015) point to a clear deficit in university students' competence with regard to the media environment. On the same note, a University of Stanford study made public in 2016 shows alarming results concerning university students' ability to gauge the credibility of online news (Wineburg et al. 2016). This is worrying, as it reveals that a university education does not seem to succeed in preparing the critical thinking of future teachers to face the digital challenge, something that is vital to forming part of a citizenry that is critical, responsible, and participatory in a democratic society. Teachers should be the first to be trained as critical citizens so that they can, in turn, educate the following generations.

We are convinced that the key to achieving a critical citizenry in the digital age lies in education and in proper teacher training at university. We want them to be ideal readers but also critical readers who read against the text (Janks 2018, p. 96).

*2.2. A Study Conducted with Pre-Service Teachers*

In this study we analyzed some CDL skills and attitudes shown by primary education undergraduates when interpreting news appearing in the Spanish digital media about sensitive issues in our society, such as poverty, social justice, and how the media influence people's opinions. We used real news items appearing in digital media and dealing with issues that urgently need to be addressed through education.

Comber (2015) claims that poverty and social justice are some of the most important topics to be addressed by critical literacy and should lead to problematizing the class and public texts. CDL can only be developed by dealing with social problems (Berson et al. 2017; Stoddard and Marcus 2017).

We probed students' social representations (Moscovici 1993) concerning these major social problems (Evans and Saxe 1996; Legardez 2003; Pagès and Antoni 2011) and controversial issues (Hess 2008; Ho et al. 2017). We also analyzed their CDL skills and attitudes to see whether they put information in quarantine, check it against other sources, uncover the hegemonic discourses behind the text, and build their own narratives critically.

## 3. Research Objectives and Methodology

This study is part of a research project funded by the Spanish Ministry of Science, Innovation and Universities (I+D EDU2016-80145-P), and involving several Spanish universities, some of whose findings we present in advance.

The main aim of this research is to investigate the skills and attitudes of future teachers in critically analyzing information supplied in digital media: assessing the veracity, reliability, or intent of information sources on social problems; discussing the credibility of certain news and other content that is spread through the media and social networks; and finding the intention and ideology beyond the information to answer the research question: Are future teachers critical thinkers?

*3.1. Participants*

The sample of 322 students consisted of third- and fourth-year students taking a bachelor's degree in primary education between fall 2017 and spring 2018 at five of the universities involved in the research project: University of Jaén; University of Málaga; University Jaume I, in Castellón; Florida Universitaria, in Valencia; and Autonomous University of Barcelona (UAB).

*3.2. Sources and Analysis of Data*

Data were collected from a series of open-ended activities and questions, which were undertaken individually within a period of 45 min and which invited the students to reflect and reveal to what extent they are critical readers of digital media and how they construct their own narrative about the social problems being dealt with. For every question, students could refer to any type of digital source to verify the information, fill in gaps, or search for new sources that might refute it.

The tool used was an online questionnaire, which allowed us to obtain certain quantitative and qualitative data with the aim of appraising the students' ability to analyze different information items and formulate critical discourses on controversial, value-ridden topics. We analyzed the discourses constructed by the students on the basis of two of the questionnaire activities, which asked them to critically analyze news items from online outlets, the content of which was problematic and needed to be verified and interpreted. In the first activity the students reflected on two news items dealing with poverty and social exclusion in Spain. In the second activity they interpreted a set of emotive, value-ridden news items that were slanted in their presentation and which they could check up on.

In order to analyze the information obtained from the questionnaire, based on the research conducted by Wineburg et al. (2016), we developed a rubric for analyzing the narratives in terms of whether these were critical or not, adopting three categories: beginning, emerging, and mastery. With this rubric, we not only consider students' informational and literacy skills but also their critical

thinking attitudes (Castellví 2019) to not only read between the lines but also beyond the lines (Gray 1960) uncovering the intention and ideology hidden in the discourse. Using the rubric, we categorize discourses that don't show any of the critical thinking skills and attitudes mentioned above as beginning, discourses that show some of them as emerging, whilst discourses that evidence the mobilization of informational, digital and literacy skills, and attitudes (Castellví 2019) and uncover the ideology hidden in the information are categorized as mastery.

## 4. Research Findings

### 4.1. Analysis of the First Activity

In the first activity (Figure 1) the students were asked to critically analyze two news items on child poverty and its impact on children's education, taking into account that, although the above is the central topic of the items, these actually contained surreptitious advertising for a bank and therefore had hidden commercial interests. The news items the students were given are the following:

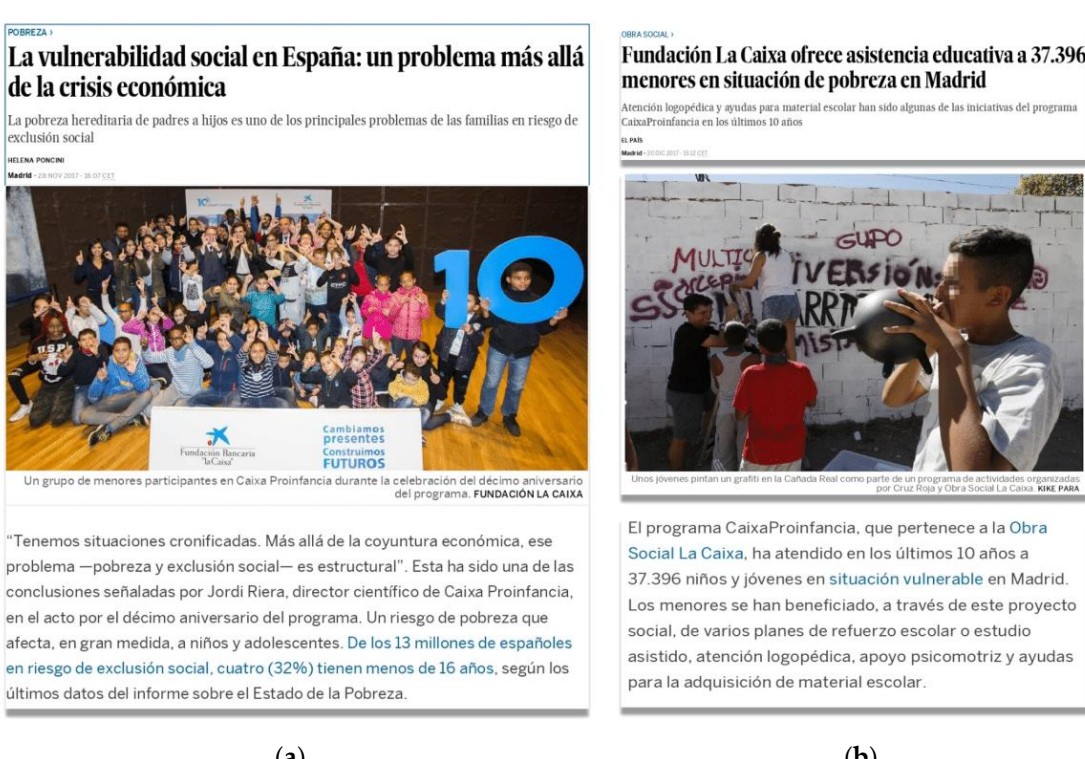

(**a**)                    (**b**)

**Figure 1.** (**a**) Social vulnerability in Spain: a problem beyond the economic crisis. (**b**) La Caixa Foundation offers educative assistance to 37,396 minors in a situation of poverty in Madrid. Pieces of news presented in the activity 1.

Below is a copy of the rubric (Table 1) created to analyze the results of this activity.

**Table 1.** Rubric utilized to analyze results of activity 1.

| Beginning | Emerging | Mastery |
|---|---|---|
| They describe the pieces of news, the social issues presented, and the main protagonists. They don't specify the interests of the bank entity or the agenda of the newspaper. | They analyze the poverty as a social issue and retrieve data from the pieces of news. They pinpoint the prominence of the bank entity in the discourse. They don't specify the interests of the bank entity or the agenda of the newspaper. | They analyze the poverty as a social issue and retrieve data from the pieces of news. They pinpoint the prominence of the bank entity in the discourse. They uncover the interests of the bank entity and the agenda of the newspaper. |

The results obtained show that the discourses of the primary education undergraduates from the participating universities fall mainly within the category of "beginning". Of the future teachers who took part in the study, 65% (Figure 2) engaged superficially with the news items and did not discover their hidden intent. In this group can be found arguments like those of Martha and Peter (all names are fictional but reflect the gender identity given in the questionnaire):

> Martha: "In the first news item you can see how more and more young people are suffering from social exclusion and poverty. And the second one is about the world we live in, where more and more people are struggling to get by each day on the little they have".

> Peter: "The first news item talks about how most child poverty is inherited from the parents, so it's tied up with problems in getting a good education. The second one gives some solutions to put an end to that inherited poverty and educate people so that they can find a better future".

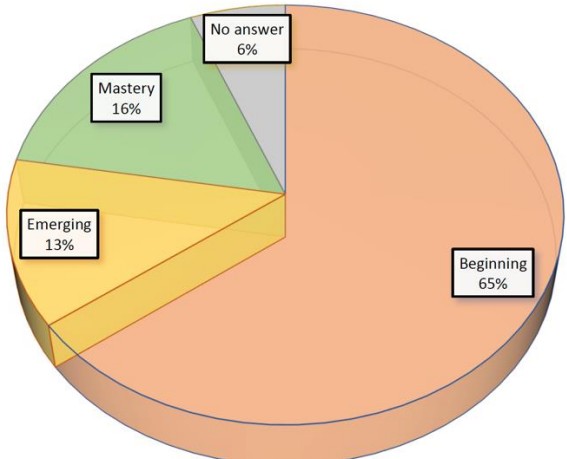

**Figure 2.** Results obtained from the analysis of activity 1.

Meanwhile, 13% of the students put together an "emerging" discourse containing some elements of reflective or judicious thinking, but not enough for the discourse to be seen to have the fundamental components of critical thinking. The answers below were given by Jessica and Ann. Jessica sees that the news outlet's focus is not on young people but on the bank, while Ann realizes that the news item tries to put the bank in a good light.

> Jessica: "The first item emphasizes the human interest part, as it focuses on children at risk of social exclusion [...]. The second item is more about highlighting the materials and projects that La Caixa has offered to these children. It also gives a complete list of the organizations that have collaborated in this initiative, forgetting about the central characters: the children".

> Ann: "My interpretation is that the news item gives some very striking data on poverty in Spain but with the underlying message that La Caixa is closely involved in a project to combat that situation".

Discourses placed in the "mastery" category are also in a minority: 16% of the total (Figure 2). These display all the elements required for them to be classified as critical discourses, as they spot the advertising intent behind the news and criticize financial entities' role in creating social inequality. They also criticize the news outlet for adopting the format of a news item for what is in fact an advertisement. These elements can be seen in the answer given by Carol:

Carol: "[...] we should take notice of who is doing the talking: the interviewee in this case is Jordi Riera, scientific director of Caja Proinfancia. [...] Very often, banks give loans or mortgages to families and if they can't repay them, they end up even more in debt or get evicted. Debts can even be inherited by the children. In this news report though, they come out as heroes, saying that the foundation has helped a lot of children in programs dealing with remedial teaching, leisure activities, hygiene, or nutrition, when in fact they're the same banks that cause families to be poor. [...] It sort of looks like the press are mounting an advertising campaign for them, which is really underhand".

The results of the first activity show that the majority of the pre-service teachers who participated in the research did not check the information presented against other sources and made a superficial interpretation of the piece of news. Furthermore, 65% did not show the skills and attitudes necessary to critically analyze the information presented and discover its intention and the bank's agenda.

*4.2. Analysis of the Second Activity*

In the second activity (Figure 3), the aspiring teachers had to show whether they could read the news from a critical standpoint and cross-check it against other media outlets. The social problem that they addressed in this second activity was not the piece of news they are shown, but the power of the media to manipulate and drive the opinion of people. Aspiring teachers were asked to comment on two successive news items, first on the initial story and then on the follow-up piece, to see if they were capable of changing their minds in light of new information. In the first news story, set in a hotel, some parents are said to have shut their small child in a safe. In the second story, this information is modified, as the parents were not there, and the children were playing hide-and-seek. The idea is for them to revise their initial opinion in light of the new information, if they had not already cross-checked the story by referring to other media outlets or other sources.

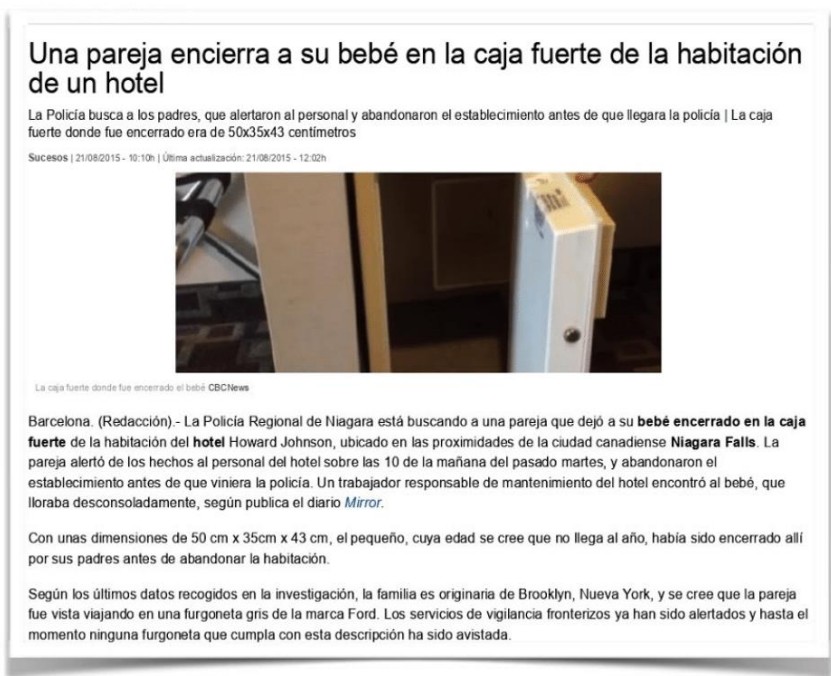

**Figure 3.** A couple locks their toddler in the safe of their hotel's room. Piece of news presented in the activity 2.

To be classified as "mastery", it is important for the students to go beyond what the news stories provide and critically appraise not only the participants' attitudes, beginning with the parents, but

also the particular intentions of the media outlets pushing the story. The news stories presented to the students were the following:

As in the previous activity, a rubric was drawn up to classify the students' answers, as shown in Table 2.

**Table 2.** Rubric utilized to analyze results of activity 2.

| Beginning | Emerging | Mastery |
|---|---|---|
| They don't contrast the piece of news with other information sources. They describe the pieces of news, the social issues presented, and the main protagonists. They judge the parents but not the media or they own uncritical attitude. | They don't contrast the piece of news with other information sources. They describe the pieces of news, the social issues presented, and the main protagonists. They judge the parents and the media, and their own uncritical attitude. | They contrast the piece of news with other information sources. They describe the pieces of news, the social issues presented, and the main protagonists. They judge the parents but also the role of the media in society. |

The findings of this activity follow the same pattern as those of the first one but deviate slightly. Those who produced discourses placed in the "beginning" category made up 33% of the total (Figure 4). These discourses are for the most part highly emotive and are not built up from coherent, reasoned argument, even when the students have access to further information that could make them revise their opinions. The information is not called into question but assumed to be valid without consulting online media as suggested from the beginning. The answers given by Frank and Kevin are two examples of this.

Frank: "My opinion is that I hope the baby is taken out of these parents' custody. If it was up to me, life imprisonment".

Kevin: "The decision to have kids is taken together and there are ways to avoid having them, like contraceptives. In some cases, people have a baby and are then shameless enough to have it adopted. Other times, we get a situation like this, which I can only see one solution for: castration for both parties and prison. If a couple are irresponsible enough to have a child they don't want, like I said, you've got adoption. But the kid shouldn't have to die through the fault of thoughtless, selfish parents".

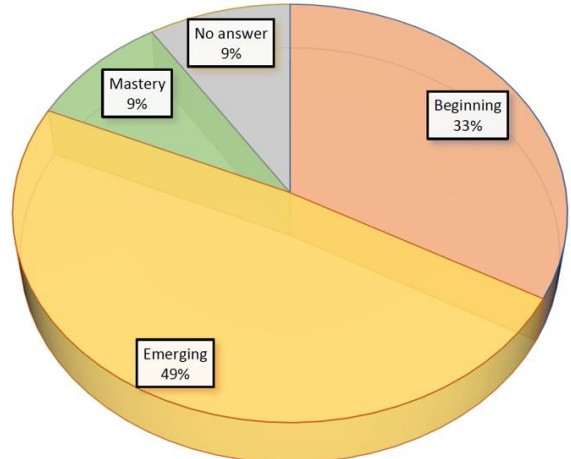

**Figure 4.** Results obtained from the analysis of activity 2.

Discourses classified as "emerging" made up the majority in this case, reaching 49% of the total number of discourses in this activity. Below are the answers given by two students, Susan and Adam,

who at first react viscerally to the news story but, on reading the second story, are able to reflect critically on the media.

> Susan (commentary on the first news story): "I think it's twisted for parents to be capable of abandoning the one they should love most in the world like that [...]".

> Susan (commentary on the second news story): "The media can be very sensationalist, and they tell stories the way they think they'll sell the best, even if it's far from the truth".

> Adam (commentary on the first news story): "I think the parents should get a big fine and go to prison for a while, because there's no excuse for what they've done".

> Adam (commentary on the second news story): "I think the media write too fast sometimes, without knowing properly what has happened, I mean, they hear rumors and assume they're right straight away".

Only 9% of the discourses could be classified as "mastery" or critical: produced by students who query the veracity of the information at all times, cross-checking it and reflecting critically on the influence of the media, on how information is manipulated, and on the importance of cross-checking against other sources. This can be seen in the answer given by Sarah.

> Sarah: "They're directly accusing the parents of something really heavy that's not ethical and doesn't make sense to do to anybody, especially your children. They haven't done any research before writing this story and they're making judgements. [...] People should be able to read and learn about what's going on and being talked about in the world, but we should be able to analyze and criticize what we're reading or what we're being told. We should be able to judge what we're being told or at least query it".

The results of the second activity are consistent with the results obtained in the first activity, as 33% of pre-service teachers did not show skills and attitudes needed to check information and critically analyze the social problem presented. It is remarkable how "beginning" and "emerging" narratives evoke emotions regarding the issue presented. However, the "emerging" ones are capable of reflecting on the importance of checking information and the influence of the media in society in that they are asked to compare the first piece of news that is biased, with the second one that clarifies the case.

## 5. Discussion and Conclusions

When analyzing the students' responses in the two activities, what strikes us most is how low they scored on capacity for critical thinking when dealing with social problems. The results obtained are in line with those of the Stanford History education group (McGrew et al. 2017; Wineburg et al. 2016), whose report goes so far as to call them devastating. The results are also in line with those obtained in other research studies with pre-service teachers (Santisteban et al. 2020) and with secondary school students (Castellví et al. 2020b). These low scores could be attributed to the fact that (1) education, at any level, mostly provides students with uncritical skills that have the appearance of being politically neutral; (2) the spaces of socialization such as the Internet or the media could have bigger impact on people's opinion than formal and non-formal education; (3) political discourses evoke emotions (Castellví et al. 2019), which are powerful tools to spread ideas without making a critical reflection about them.

The data obtained in the two activities point to a very small minority of students formulating critical discourses at the "mastery" level: below 20% in both activities. This percentage is worrying, especially in light of the fact that these students will have the responsibility of teaching the critical citizenry of the future. If they themselves are not capable of questioning information in a world dominated by multiple digital media, they are even less likely to be able to help their pupils to question power relations and work to promote social justice and democracy.

Most of the students put together discourses that were placed in the "beginning" category in the first activity, and as many as 65% were unable to spot a news story with an underlying commercial intent. For the second activity the number was 33%. These percentages seem very high for students in the second half of their university studies, who should by now be used to cross-checking information to establish its reliability. This leads us to the following questions. How can critical thinking be taught in pre-university education? How should it be taught at university? In this new context it seems appropriate to reconsider the role of future teachers within the education system (Kendall and McDougall 2012) and the objectives of initial training programs.

In the second activity the category with the largest number of students (49%) is the "emerging" category, while, in the first activity, 13% fall into this category. We could initially have expected these students to be well used to using digital resources, interpreting information they can access easily and weighing up the veracity of what appears on the multiple screens they use. But the study's findings indicate, rather, that they do not have these abilities or do not put them into practice.

According to Gutierrez and Tyner (2012), media education in the different stages of education should not be limited to developing a merely instrumental digital competence, neglecting attitudes and values. However, our research suggests that, while attaining technical mastery of the medium, few students show more than a negligible willingness to take a step further and critically interpret information. Our research also shows that there is still much work to be done with university students regarding the social problems at the core of the activities and regarding critical reading of the media, as seen also in other studies (Alonso et al. 2015; Wineburg et al. 2016).

As we see it, beyond the question of whether a particular story is interpreted one way or another, aspiring teachers must take a clear stance on human rights and democratic values. By teaching how to make use of digital resources, we should be producing responsible, cultivated, critical citizens (Area-Moreira and Pessoa 2012), who are fully aware of their social function as teachers. It is clear that teacher-training programs need to be rethought with the aim of incorporating critical media literacy. Additionally, they should include the study of social problems or controversial topics that can call into question how we approach the media and interpret information.

We are convinced of the need for CDL in citizenship education, to prepare citizens for democratic participation. We propose working with social problems, as an essential education tool in today's democracy, to help develop critical thinking (Santisteban 2019). One question we could ask ourselves is whether someone is educated if they are not capable of distinguishing between manipulation and information. We could also ask ourselves whether we can assure the future of democracy without critical, participative, and responsible citizens. These should be our main challenges as teachers.

**Author Contributions:** Conceptualization, J.C. and A.S.; Data curation, J.C.; Formal analysis, J.C.; Funding acquisition, A.S.; Investigation, J.C., M.-C.D.-B. and A.S.; Methodology, A.S.; Project administration, A.S.; Resources, A.S.; Software, J.C.; Supervision, M.-C.D.-B.; Validation, J.C., M.-C.D.-B. and A.S.; Visualization, J.C.; Writing—original draft, J.C. and M.-C.D.-B.; Writing—review & editing, J.C. and M.-C.D.-B. All authors have read and agreed to the published version of the manuscript.

**Funding:** This publication is part of the R & D Project 'Teach and learn to interpret contemporary problems and conflicts. What do the Social Sciences contribute to the formation of a critical global citizenship?', funded by the Spanish Ministry of Science, Innovation and Universities (EDU2016-80145-P), whose main researcher is Professor Antoni Santisteban (Autonomous University of Barcelona).

**Conflicts of Interest:** The authors declare no conflict of interest.

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
