# Peer review of "Pre-Service Teachers’ Critical Digital Literacy Skills and Attitudes to Address Social Problems"

_socsci, doi:10.3390/socsci9080134_

Round 1

Reviewer 1 Report

This study explores a timely and important issue for teacher education and critical pedagogy, which is to what extent preservice teachers are critical readers of digital media. Building from the idea that critical thinking is not developed in abstract but through the analysis of current social problems, the author/s use an appropriate tool for data collection based on analyzing up-to-date news items. The results, although not surprising, provide relevant nuances of what particular dimensions of critical digital literacy should be paid more attention to in future research and teacher education practice. For these reasons, I think that this paper is valuable to be published in Social Sciences.

In order to provide a clearer view of the study to the readers, I would suggest the authors to introduce the following clarifications about the conceptual framework and the methods used. These would strengthen, in my opinion, the findings of the study:

  • The author/s analyze preservice teachers’ critical digital literacy skills and attitudes when interpreting certain news items. However, which are these skills and attitudes? They mention some throughout the paper: analyzing multiple multimodal texts, reflecting on information, uncovering hegemonic discourses, etc. Yet, considering the importance of these concepts for the study, they need to be more explicitly addressed in the paper.
  • With regards to the data collection method, it would be very helpful if the author/s provide more detail on the questions used in the questionnaire and explain the rationale behind the selection of the news items (in particular, it needs to be further explained why the second news item was chosen, as it does not refer to a social problem…).
  • The author/s should provide more detailed information on the instrument used for the analysis of the data (the rubrics) in the methodological section. This would be clearer if they make explicit what they mean by critical literacy skills and attitudes and which skills/attitudes correspond to each category of the rubrics (beginner, emerging, and mastery). As it is stated now, it seems that the criteria vary depending on the data collection method used (activity 1 or 2).

Finally, I recommend the author/s to expand a bit more on the ‘so what’ of their work. They conclude that “that teacher training programs need to be rethought with the aim of incorporating critical media literacy” and “include the study of social problems or controversial topics”, but this conclusion does not add anything new to the field. I feel they could be more specific about their contributions and deepen a bit more on the implications of their results.

Author Response

Dear revisor,

We appreciate your comments and suggestions and we think they were very helpful to strengthen our manuscript.  

+The author/s analyze preservice teachers’ critical digital literacy skills and attitudes when interpreting certain news items. However, which are these skills and attitudes? The author/s should provide more detailed information on the instrument used for the analysis of the data (the rubrics) in the methodological section. 

We explained better the critical thinking skills and attitudes we refer to in the study. Specially, we went in depth with their relations with the rubrics we utilized to analyze pre-service teachers’ discourses.

+With regards to the data collection method, it would be very helpful if the author/s provide more detail on the questions used in the questionnaire and explain the rationale behind the selection of the news items

We agree that this part was not strong enough. We extended the explanation of the selection criteria for the news, focusing on the second piece of news and the social problem we address there.

Reviewer 2 Report

The article deals with a relevant, current and necessary issue for educaton in complex and uncertain tiems sucha as the ones we are livin in. In this context, social science teacher training faces the challenge of undertaking research on the impact of the digital world on the perception of social facts and processes. 

The research has a wide sample to know the critical thinking skills of future teachers and their ability to analyze the intentionality and ideology of the information collected in digital media.

The analysis of results is broad and detailed, reaching relevant conclusions, among whixh we highlight the need to deepen media education focused on attitudes an values.

Author Response

Dear revisor,

We appreciate your comments and suggestions. We entirely agree with you with these reflections. We will go further on this research and how CDL can make us rethink Citizenship Education at all levels. 

Reviewer 3 Report

The aim of this paper is to explore Spanish teacher students capacity for constructing critical discourses and the importance of such skills to prepare citizens for democratic participation. In this paper they ask teacher students to work with social problems as an essential tool to increase critical thinking and a deeper understanding of democracy today. Empirical findings are presented and analyzed with transparency and key findings are of high relevance internationally. Interesting and important paper about skills to prepare citizens, teachers and learners, for democratic participation. The strength of this paper is the empirical data, the description of analyzing tools and the number of participants. Theories and relevant research are also presented and discussed well as well as a conclusion part. Generally, well written academic paper. This paper can still improve if the research question come out more explicit, more theories about critical thinking (emphasized more on Freire) and discussed why majority of the students are not able to construct critical discourses. Society, education, contextualizing learning conditions at the Universities in Spain, which subjects could increase their skill etc. Can Western neoliberal ideology be an explanation for young people/students lack of critical thinking? (Bauman, Giddens etc). Line 158 can continue from line 157 since this is within the same quote. The paper is qualified for publication, but the author(s) should consider to strengthen the discussion part with relevant theories and explicit highlight the research question.

Author Response

Dear revisor,

We appreciate your comments and suggestions and we think they were very helpful to strengthen our manuscript.  

+This paper can still improve if the research question come out more explicit.

We made it more explicit in the methodology section.

+More theories about critical thinking (emphasized more on Freire).

We strongly agree with this suggestion. We added a paragraph presenting Freire’s work and the importance of his critical pedagogy to CDL skills and attitudes.

+Discussed why majority of the students are not able to construct critical discourses. 

We discussed on this topic in the last section. Despite there isn’t a clear answer to this question, we pointed out some reflections and evidences that shade light on this issue.